# Assortative mating and within-spouse pair comparisons

**Laurence J. Howe** [1,2] *, **Thomas Battram** [1,2], **Tim T. Morris** [1,2], **Fernando P. Hartwig** [1,3], **Gibran Hemani** [1,2], **Neil M. Davies** [1,2,4], **George Davey Smith** [1,2]

**1** Medical Research Council Integrative Epidemiology Unit, Population Health Sciences, University of Bristol, Bristol, United Kingdom, **2** Population Health Sciences, Bristol Medical School, University of Bristol, Bristol, United Kingdom, **3** Postgraduate Program in Epidemiology, Federal University of Pelotas, Pelotas, Brazil, **4** K.G. Jebsen Center for Genetic Epidemiology, Department of Public Health and Nursing, Norwegian University of Science and Technology, Trondheim, Norway

* laurence.howe@bristol.ac.uk

**Data Availability Statement:** In this study, we used individual participant data from UK Biobank. Interested researchers would be able to obtain the same data-set from UK Biobank. Reference: https://www.nature.com/articles/s41586-018-

## Abstract

Spousal comparisons have been proposed as a design that can both reduce confounding and estimate effects of the shared adulthood environment. However, assortative mating, the process by which individuals select phenotypically (dis)similar mates, could distort associations when comparing spouses. We evaluated the use of spousal comparisons, as in the within-spouse pair (WSP) model, for aetiological research such as genetic association studies. We demonstrated that the WSP model can reduce confounding but may be susceptible to collider bias arising from conditioning on assorted spouse pairs. Analyses using UK Biobank spouse pairs found that WSP genetic association estimates were smaller than estimates from random pairs for height, educational attainment, and BMI variants. Within-sibling pair estimates, robust to demographic and parental effects, were also smaller than random pair estimates for height and educational attainment, but not for BMI. WSP models, like other within-family models, may reduce confounding from demographic factors in genetic association estimates, and so could be useful for triangulating evidence across study designs to assess the robustness of findings. However, WSP estimates should be interpreted with caution due to potential collider bias.

## Author summary

There is growing evidence that genome-wide association studies capture associations relating to environmental factors, such as indirect effects from parental genotypes. Within-family models such as sibling comparisons can be used to disentangles these different sources of association but are limited by the paucity of sibling data in large biobanks. Within-spouse pair models are a potentially tractable model because spouses share environmental factors in adulthood and may also share early-life environmental factors. Here, we evaluated the application of within-spouse models in genetic association studies, specifically considering assortative mating, a phenomenon whereby individuals may select a phenotypically similar partner. We found that within-spouse pair models can detect

0579-z Contact: access@ukbiobank.ac.uk.
Relevant code for simulation models is available at
the following repository https://github.com/
LaurenceHowe/Between-spouse.

**Funding:** LJH, TB, TTM, GH, NMD and GDS are
members of MRC Integrative Epidemiology Unit
which is supported by the Medical Research
Council (MRC) [MC_UU_00011/1] and the
University of Bristol (principal investigator: GDS).
NMD is supported by The Economics and Social
Research Council (ESRC) via a Future Research
Leaders grant [ES/N000757/1], a Norwegian
Research Council Grant number 295989 and by the
Health Foundation's Efficiency Research
Programme (Award 807293). The funders had no
role in study design, data collection and analysis,
decision to publish, or preparation of the
manuscript.

**Competing interests:** The authors declare no
competing interests.

genuine confounding in genetic association estimates but are potentially susceptible to collider bias induced by comparing assorted pairs. Within-spouse pair estimates could be useful when combining evidence from different study designs.

## Introduction

Within-sibship models have been widely used in genetic association studies for many decades [1–6]. Genotypic differences between siblings are the consequence of random segregation at meiosis, rather than parental or ancestral differences, and so within-sibship genetic association models control for demography (assortative mating, population stratification) and indirect genetic effects of parents [4,5,7]. However, there is a paucity of genetic data from siblings with limited availability for many phenotypes. Furthermore, while siblings are matched on the early-life environment, their environments in adulthood, when many phenotypes are measured, may differ.

In contrast, spouses may have different early-life environments but are likely to share an environment for much of adulthood while cohabiting [8]. This may act to increase phenotypic similarity, such as for behavioural (e.g., physical activity and alcohol use) or personality traits [9,10]. The shared adulthood environment between spouses has prompted their use in a variety of contexts in genetic and epidemiological research using a model that we refer to as the "within-spouse pair" (WSP) model. The WSP model involves modelling the similarities and differences of spouses, either by analysing the differences between each pair or by modelling spousal relationships as a covariate in a fixed-effect model. For example, previous studies have used the WSP model to estimate phenotypic variance explained by the shared adulthood environment [11–17]. The WSP model has also been proposed as an approach to reduce confounding in aetiological research, with environmental confounders likely to be strongly correlated between spouses [18]. Here, we describe the strengths and limitations of within-spouse (WSP) designs for genetic studies.

A caveat of the WSP model is that spousal similarities are not just consequences of sharing an adulthood environment. There is evidence that for some phenotypes, spouses do not become much more similar during a relationship [19]. Another cause of spousal similarities is assortative mating–a phenomenon where humans are generally more likely to select a phenotypically similar [9,10,20–26] or, in some instances [27], dissimilar [28,29] mate. For example, height and years in schooling are often fixed prior to partnership formation, suggesting that spousal similarities for these phenotypes reflect assortment rather than effects of the shared adulthood environment. Furthermore, geographical, ancestral, and cultural factors often have strong influences on both phenotypic variation and partner selection patterns, as illustrated by the ancestral similarities of spouses [30]. Therefore, some degree of spousal phenotypic similarities is likely to be explained by spousal assortment on factors not typically defined as phenotypes, such as place of birth or religion.

The WSP model may be susceptible to collider bias, which can occur when conditioning on a variable which is influenced by two or more upstream factors. Collider bias can induce spurious associations between these factors where the collider variable is conditioned on in analysis either by analytical model design or sample selection. For example, if a school grants scholarships to either individuals who are exceptional at sport or exceptional academically, then sporting and academic ability will be negatively correlated amongst individuals with scholarships. Similarly, spousal samples by definition condition on spousal compatibility, a pairwise measure of how likely two individuals are to enter a relationship. If several phenotypes influence

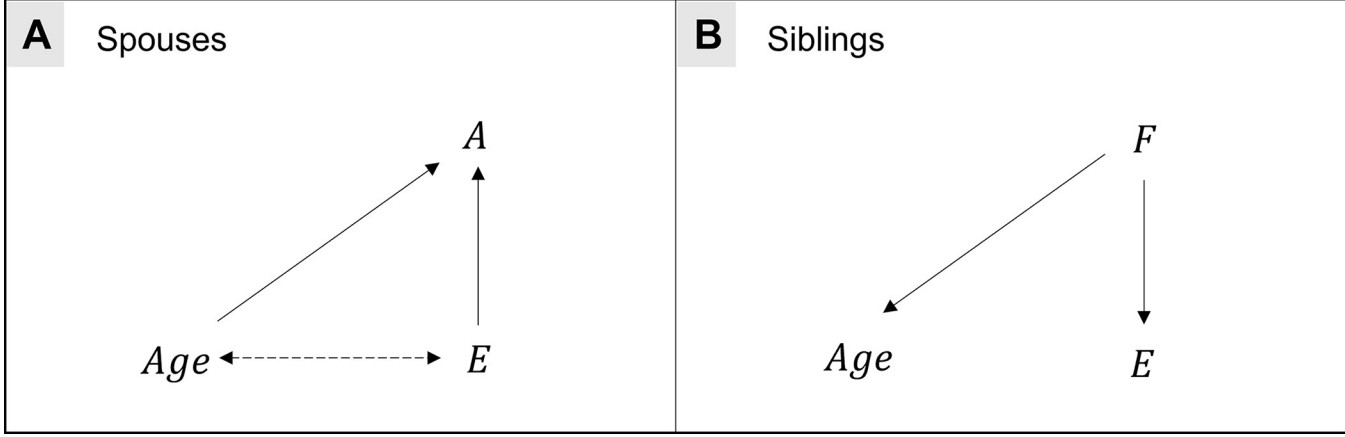

**Fig 1. Causal diagram illustrating collider bias in within-spouse pair and within-sibship models.** A) To illustrate collider bias in the context of spouses, consider a model with age and educational attainment (*E*) which are assumed here to be independent. Assuming that spouses assort on similarities for age and education, it follows that spousal assortment (*A*) is a common effect of age and education similarities. In a within-spouse pair model, adjusting or accounting for *A* would induce associations between age and education similarities. For example, if a spouse-pair have a large difference in age, then they must be similar for education. B) Contrastingly, for a within-sibship model, it is less plausible that age and education influence the sibling's family *F*, as age and education are post-birth phenotypes. Therefore, adjusting for *F* is unlikely to induce an association between age and education.

spousal compatibility, then collider bias could potentially arise in the WSP model [31–33]. For example, if similarities for age and educational attainment influence compatibility then spouses with larger age differences are more likely to have similar educational attainment (**Fig 1**). Previous spousal studies have acknowledged assortative mating, but whether assortment could distort WSP comparisons has not been investigated in detail. For example, the possibility of collider bias has been little discussed. We aimed to investigate the utility of the WSP model in genetic epidemiology and assess its robustness to collider bias.

We used causal diagrams (allowing double-headed arrows signifying correlated variables that may be influenced by variables outside the model [34]) and simulated data to illustrate two important characteristics of the WSP model. First, the WSP model can reduce confounding if spouses are correlated for confounders. Second, the WSP model is susceptible to collider bias induced by conditioning on spousal compatibility. We then applied the WSP model using 47,435 spouse-pairs in UK Biobank [35] to estimate associations between genetic variants and phenotypes (e.g. height). We then estimated effect size shrinkage (% decrease) in the WSP estimates compared to within-pair estimates from random non-assorted pairs, which were derived by reordering the spouse-pair sample. For comparison, we also estimated within-sibship shrinkage using 19,523 sibling pairs from UK Biobank. Finally, as a negative control analysis, we used the WSP model to estimate the effects of age on systolic blood pressure (SBP) and coronary artery disease (CAD).

## Results

### Within-spouse pair model: Assortative mating, spousal correlations and collider bias

Here, we present results from simulations evaluating the WSP model under assortative mating. In the first simulation model (A), the relationship between an exposure and an outcome is confounded by an unmeasured factor. Spouses are positively correlated for the unmeasured confounder, either because of assortative mating or because of shared environmental factors during cohabitation (**Fig 2A**). Simulations demonstrated that under this model, WSP

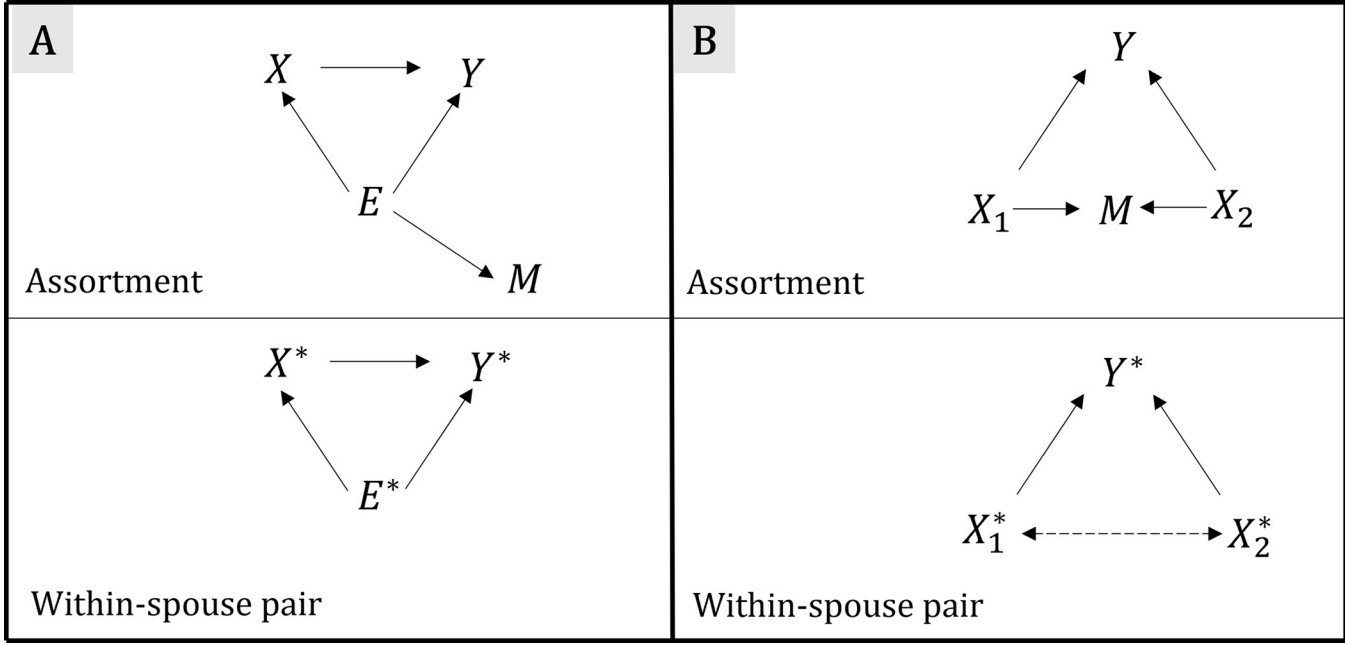

**Fig 2. Causal diagrams of simulated models for assortative mating, spousal correlations, and collider bias.** The WSP design uses pairwise spousal differences (e.g. $X_{M1}-X_{F1}$ & $Y_{M1}-Y_{F1}$) in regression models, fitting each spouse pair as a single observation. A) *Within-spouse pair: spousal correlations for confounders.* Exposure $X$; Outcome $Y$; Unmeasured confounder $E$; Spousal assortment $A$; WSP exposure $X^*$ ($X^* = X_M-X_F$); WSP outcome $Y^*$ ($Y^* = Y_M-Y_F$); WSP environmental confounder (the non-shared portion of the set of confounders) $E^*$ ($E^* = E_M-E_F$). This figure illustrates the effect of an exposure on an outcome in the presence of an unmeasured confounder. Here, spousal pairing is determined by an assortment variable correlated with the confounder (indicated by A, a child of the confounder E). It follows that the value of spouses' confounders will be correlated. In this example, a WSP model will reduce simulated bias in the estimate of the effect of X on Y (S1 Fig). Here we assume that spousal correlations for the confounder reflect assortment but in practice they could also relate to the shared spousal environment. B) *Within-spouse pair: assortative mating and collider bias.* Exposures $X_1$ $X_2$; Outcome $Y$; Spousal assortment $A$; WSP exposures $X_1^*$, $X_2^*$ ($X_i^* = X_{iM} - X_{iF}$); WSP outcome $Y^*$ ($Y^* = Y_M-Y_F$). This figure illustrates the effect of an exposure on an outcome when two, otherwise independent exposures influence both the outcome and spousal assortment. It follows that associations will be present in the WSP model between the two exposures, which will distort the WSP estimated effect of the exposure on the outcome. We quantify the effect of potential collider bias in the WSP model at different levels of assortment on the two exposures. Dashed lines indicate associations induced by spousal assortment.

estimates of the effect of the exposure on the outcome are less biased and converge to the simulated unbiased estimate as the spousal correlation for the confounder tends to 1 (**S1 Fig** and **S1 Table**).

In the second simulation model (B), two independent exposures influence an outcome. These exposures could be two phenotypes, a phenotype and a genetic score, or two independent genetic scores such as height genetic scores constructed from odd and even chromosomes. Since there is assortment on the two exposures, assortment acts as a collider, which induces associations between variables that would otherwise be independent in the population. For example, in the WSP model, positive assortment on height and educational attainment phenotypes could induce a negative correlation between height and educational attainment genetic scores. The strength of this spurious correlation will depend on the underlying data generating process and the degree of assortment on the exposures. Indeed, assortment on height across multiple generations has resulted in positive correlations between height increasing genetic variants on different chromosomes [22]. These correlations could lead to bias in WSP estimates of either exposure on the outcome. However, WSP estimates will only be affected by collider bias if both exposures influence the outcome and if the effects of the exposures on the collider are not perfectly multiplicative (**Fig 2B**).

Simulations showed that the degree of bias in the effect estimate is a function of the degree of assortment on the two exposures, with more bias when spouses assort strongly on both

traits. For example, under this model and using plausible assortment estimates for educational attainment (spousal phenotypic correlation: 0.5) and height (spousal phenotypic correlation: 0.2) [25], the expected bias would be around 13% when estimating the effect of education on a trait which is also influenced by height. If both exposures have the same direction of effect on the outcome and assortment, then the WSP estimate would be biased downwards (**S1 Fig and S2 Table**).

## Empirical analyses using spouse pairs in UK Biobank

**Within-spouse pair: genetic and phenotypic associations.**    Although the WSP model is susceptible to collider bias, the model could be useful in re-calibrating associations that might be biased due to confounding. Genetic data are particularly useful for evaluating aetiological models because genotypes are measured accurately and fixed from conception, largely removing the possibility of reverse causation. Here we aimed to evaluate if WSP genetic association estimates are less confounded than estimates from population studies of unrelated individuals.

Estimates of genetic associations using unrelated individuals can be distorted by demography (e.g. assortative mating, fine-scale population structure) and indirect effects of parents [6,7,36]. WSP estimates may be less affected by these sources of association, particularly population structure, because of environmental and ancestral similarities between spouses [30].

Using 47,435 spouse-pairs from UK Biobank (**S3 Table and S2, S3 and S4 Figs**), previously derived using household sharing information [24], we first explored the extent to which spouse and sibling pairs are correlated for the first 10 principal components and birth coordinates (north-south, east-west) to inform the extent of pairwise spousal ancestral similarities. Spouse pairs were correlated for both birth coordinates and the first 10 principal components with correlations ranging from 0.10 for PC6 to 0.32 for PC4 across the principal components and strong correlations observed for both north-south (0.62; 95% C.I. 0.61, 0.62) and east-west (0.46; 95% C.I. 0.46, 0.47) birth coordinates. As expected, sibling pairs were very strongly correlated for birth coordinates and the first 10 principal components with correlations ranging from 0.74 for PC10 to 0.98 for PC4 (**S4 Table**). The spousal correlations for birth coordinates and principal components illustrate how assortative mating and social homogamy induce ancestral similarities between spouses.

We then estimated the effects of genetic variants on six different traits using the WSP design and between unrelated non-spouse pairs. We then estimated the shrinkage (% attenuation) from the non-spouse pair genetic association estimates to the WSP estimates. For comparison, we also applied the same approach to a sample of 19,523 sibling pairs (i.e. within-sibship model). Within-sibship models are a gold standard within family design for estimating genetic associations because they control for demographic and parental effects [4,6,36,37]. Comparisons between WSP and within-sibship shrinkage estimates would provide insight into the accuracy of WSP genetic association estimates.

We found strong evidence of smaller effect sizes in the WSP model for height (shrinkage: 19%; 95% CI 17%, 22%), educational attainment (shrinkage: 72%; 95% CI 64%, 79%) and BMI (shrinkage: 16%; 95% CI 6%, 25%). Contrastingly, there was limited evidence of shrinkage for SBP and CAD variants. The within-sibship analysis provided strong evidence of shrinkage for height (shrinkage: 15%; 95% C.I. 11%, 20%) and educational attainment (shrinkage: 53%; 95% C.I. 35%, 71%) variants, but limited evidence for BMI variants (shrinkage: 5%; 95% -12%, 22%). WSP shrinkage estimates were generally higher than within-sibship estimates, but imprecision prevented stronger conclusions regarding heterogeneity. Including principal components in the random-pair models did not greatly affect results except in the alcohol analysis where there was only evidence for shrinkage in the unadjusted models. This suggests

**Table 1. Estimates of genetic association shrinkage from within-spouse pair and within-sibship models.**

| Phenotype | Number of SNPs | Covariates | Within-spouse pair shrinkage: % (95% C.I.) | Within-sibship shrinkage: % (95% C.I.) | Heterogeneity P for spouse and sibling shrinkage estimates |
|---|---|---|---|---|---|
| Height | 381 | No PC | 19% (17%, 22%) | 15% (11%, 20%) | 0.18 |
| | | PC1-10 | 17% (14%, 20%) | 13% (8%, 18%) | 0.19 |
| Educational attainment | 69 | No PC | 72% (64%, 79%) | 53% (35%, 71%) | 0.06 |
| | | PC1-10 | 71% (62%, 79%) | 51% (33%, 70%) | 0.06 |
| Body mass index | 68 | No PC | 16% (6%, 25%) | 5% (-12%, 22%) | 0.28 |
| | | PC1-10 | 16% (6%, 25%) | 5% (-12%, 22%) | 0.28 |
| Coronary artery disease | 41 | No PC | -4% (-23%, 15%) | -1% (-36%, 34%) | 0.90 |
| | | PC1-10 | -4% (-23%, 16%) | -1% (-35%, 34%) | 0.83 |
| Systolic blood pressure | 242 | No PC | 0% (-7%, 8%) | 5% (-7%, 18%) | 0.53 |
| | | PC1-10 | 0% (-8%, 7%) | 5% (-7%, 18%) | 0.50 |
| Alcohol consumption | 1[A] | No PC | 29% (14%, 43%) | 20% (-20%, 59%) | 0.40 |
| | | PC1-10 | 14% (-5%, 33%) | 4% (-46%, 54%) | 0.42 |

A: rs1229984 in *ADH1B*

that population stratification is unlikely to entirely explain the observed shrinkage in these estimates (**Table 1**). We note that the alcohol analysis included only a single SNP which is known to be strongly associated with population structure in UK Biobank [24].

We compared the within-sibship and WSP shrinkage estimates from this study (using UK Biobank data only) to within-sibship shrinkage estimates from a recent within-sibship GWAS of 17 cohorts [5], which included over 4x as many siblings as this study. The within-sibship shrinkage estimates from the multi-cohort GWAS were highly consistent with the within-sibship shrinkage estimates from UK Biobank only but were much more precise. The multi-cohort within-sibship shrinkage estimates were smaller than the WSP shrinkage estimates for height, BMI, and educational attainment, with non-overlapping confidence intervals, providing some evidence that WSP shrinkage is larger than within-sibship shrinkage for these phenotypes (**S5 Table**).

**Within-spouse pair: age, SBP and CAD.** As a negative control analysis, we next used the WSP model to estimate the effects of increasing age on outcomes known to be related to age (CAD and SBP), using random pair estimates for comparison. Age cannot be influenced by other phenotypes, so analyses are unlikely to be susceptible to reverse causation or confounding. However, collider bias in the WSP model with age is plausible because spousal compatibility is influenced by age similarities. For example, couples with large age differences may systematically differ to couples with smaller age differences. It follows that differences between WSP and random pair estimates (with age as the exposure) are likely to reflect collider bias. This is a similar premise to autosomal GWAS of sex, where genetic associations are likely to reflect participation bias because autosomal genetic variation cannot influence sex [38].

Pairwise age differences were found to be greater between random pairs, consistent with individuals preferring a partner of a similar age. We did not find strong evidence for differences in age effect estimates on CAD and SBP between the spouse and random pair samples suggesting that any collider bias effects are modest in this context (**Table 2**).

## Discussion

In this study, we used causal diagrams, simulations, and empirical data to evaluate the use of the WSP model in genetic epidemiology. We showed that the WSP model can account for

**Table 2. Within-spouse pair estimates of the effect of age on SBP and CAD.**

| Phenotype | Spouse-pairs (N = 47,435) | Random pairs (N = 47,435): Median estimate from 100 simulations |
|---|---|---|
| Average age difference (years); Median (Q1, Q3) | 2.0 (1.0, 4.0) | 7.0 (3.0, 13.0) |
| Systolic blood pressure (Change in mmHg per 1-year increase in age; 95% C.I.) | 0.74 (0.69, 0.80) | 0.80 (0.78, 0.83) |
| Coronary artery disease (OR per 1-year increase in age; 95% C.I.) | 1.05 (1.04, 1.05) | 1.05 (1.04, 1.05) |

All analyses were adjusted for sex of the index individual.

unmeasured confounding if spouses are correlated for the confounder but that comparing assorted spouses can induce collider bias. Using empirical data, we found evidence that genetic association estimates for height, educational attainment, and BMI shrink in the WSP model when compared to a within-pair model using random individuals.

Within-sibship models in UK Biobank, which control for demographic and parental effects [6,36], also provided evidence of shrinkage for height and educational attainment variants but not for BMI, consistent with previous studies [4,5,37]. WSP shrinkage point estimates for height, BMI and education were larger than the UK Biobank within-sibship shrinkage estimates although confidence intervals overlapped. However, there was strong statistical evidence that WSP shrinkage is greater than within-sibship shrinkage for height, BMI and educational attainment when using more precise within-sibship shrinkage estimates from a recent within-sibship GWAS [5]. The consistent evidence of shrinkage between the two models for height and education suggests that WSP models may be removing associations relating to demography.

Simulated data illustrated that if spouses assort on a confounder of the exposure and outcome, then the WSP association provides a less biased estimate of the causal effect than a conventional model unadjusted for the confounder. An example of a potential confounder in genetic association studies is ancestry, which we showed to be more correlated between spouses than for non-spouse pairs by illustrating birth coordinate and principal component correlations between spouses. However, we note that including principal components as covariates did not greatly affect shrinkage estimates except for alcohol consumption where, as noted earlier, the single variant used is known to be strongly associated with population structure [24]. The WSP shrinkage estimates being higher than the sibling estimates suggests that the shrinkage cannot be explained by adjustment for confounding alone. If the only source of shrinkage is adjustment for confounding, then WSP shrinkage estimates should be smaller than within-sibship estimates because spouse models are unlikely to fully control for demographic or family-level (e.g. parental nurture) effects. Collider bias induced by comparing assorted pairs is one potential explanation for the observed WSP shrinkage.

Collider bias could contribute to the observed shrinkage depending on the interactive model between the colliding effects and the degree of assortment. Assuming a linear additive model, collider bias is likely to shrink rather than inflate genetic associations because assortment would induce negative correlations in the WSP model between factors influencing the assorted trait in the same direction. For example, assortment could induce negative correlations between height increasing genetic variants and height increasing environmental factors in the WSP model, leading to shrinkage when estimating WSP genetic associations. This is in contrast to the population-level effects of assortative mating which inflate associations because of induced positive correlations between trait-increasing variants on different chromosomes

[22]. However, in the negative control example of age on health outcomes, we found little evidence of collider bias; within-pair effect estimates of age on CAD and SBP were consistent between spouse and non-spouse pair samples. Another potential source of WSP shrinkage is the spousal environment. The spousal environment could be influenced by individual's genotypes leading to reduced spousal phenotypic differences. For example, if an individual has high genetic liability to increased alcohol consumption this could lead to their partner consuming similar amounts of alcohol independent of their genotype.

A key implication of these analyses is that spousal similarities and differences are not necessarily random or attributable solely to the shared adulthood environment. WSP similarities are likely to reflect a combination of social homogamy, assortative mating and the shared adulthood environment. Amidst growing evidence that genetic epidemiological studies can capture effects of fine-scale population structure, parental nurture and assortative mating [4,6,39–46], there is considerable interest in using genotype data from pedigrees to more accurately estimate direct genetic effects and trait heritability as well as to explore parental effects on offspring phenotypes [11–14,39–41,44,45,47–51]. Family designs such as the transmission disequilibrium test [52] and within-sibship models are protected from many of these biases by random segregation at meiosis [53,54]. However, in contrast, inferences from spousal analyses are not as robust, thus it is important to understand and model the assortment in spousal designs. A further implication is that assortative mating is likely to contribute to the phenotypic and genetic structure of epidemiological studies. Large studies such as the UK Biobank, frequently incidentally sample participants who are partnered with another study participant [24]. The non-randomness of study participation in UK Biobank has been previously discussed as a possible cause of selection (participation) bias [31]. Our findings illustrate that assortative mating is likely to contribute to the non-random distribution of phenotypes (and genotypes) in population biobanks.

Our study has several important limitations. First, as described in our previous study [24], derived spouse-pairs were identified using household sharing information so may be susceptible to a degree of classification error with non-spouse pairs being incorrectly identified as spouses. Second, the mechanisms by which spouses jointly participate in UK Biobank may have induced selection bias into empirical analyses as these pairs could be more similar than pairs that did not jointly participate. Third, given that the exact mechanisms of assortment are not widely understood, our simulations and assumptions may not accurately capture the mechanisms underlying spousal assortment. In simulations we assumed that factors influencing assortment are independent across the population but in practice, factors influencing assortment are often correlated (e.g. height and education). Future research could use more complex simulations to evaluate models that can distinguish the effects of social homogamy, migration and measurement error. Fourth, it is important to note that educational attainment as defined by qualifications when study participants are aged over 40 will also capture individuals with degrees obtained during adulthood, suggesting that educational similarities could also plausibly relate to the shared adulthood environment.

To conclude, the WSP model can reduce confounding from environmental factors but may also be susceptible to collider bias. An empirical example using genetic associations suggested that WSP estimates may be downwardly biased. Contrastingly, WSP estimates for effects of age did not seem to be affected by collider bias. An advantage of WSP models is that they may have increased power for genetic studies relative to other within-family designs because (non-consanguineous) spouses are less likely than first degree relatives to share long segments of the genome identical by descent. The WSP model could be a complimentary orthogonal design to other within-family models when triangulating evidence from different study designs [33].

## Methods

### Data sources

**UK Biobank.  Study description**

UK Biobank is a large-scale prospective cohort study which sampled 503,325 individuals aged between 38–73 years at baseline, recruited between 2006 and 2010 from across the United Kingdom. The cohort has been described in detail previously [35,55]. For the purposes of this study, we used two subsamples of the cohort; spouse-pairs [24], and full-sibling pairs [6].

Potential spouses were estimated using household sharing information in a previous publication [24]. We started with a European subsample of UK Biobank, consisting of 463,827 individuals based on a k-means cluster analysis on the first 4 genetic principal components. We then used phenotype data to extract pairs of individuals who reported (a) living with their spouse (field ID: 6141–0.0), (b) the same length of time living in the house (field ID: 699–0.0), (c) the same number of occupants in the household (field ID: 709–0.0), (d) the same number of vehicles (field ID: 728–0.0), (e) the same accommodation type and rental status (field IDs: 670–0.0, 680–0.0), (f) identical home coordinates (rounded to the nearest km) (field IDs: 20074–0.0, 20075–0.0) and (g) are registered with the same UK Biobank recruitment centre (field ID: 54–0.0) and (h) both have available genotype data. We considered pairs with identical information across all household variables as putative spouses. When more than two individuals shared identical information (observed in 18,145 instances), then these individuals were removed. 53 closely related pairs (IBD $> 0.1$) were identified and removed using a genetic relationship matrix. We excluded 4,866 potential couples who were the same sex (9.3% of the sample) as they were deemed to be more likely to be false positives and because of possible heterogeneity in same-sex assortment patterns. The original paper identified 47,549 male-female pairs believed to be cohabitating spouses. In this study, we used an updated version of the genetic data after removing individuals who had opted out of the study resulting in a slightly reduced sample of 47,435 complete pairs.

Full-sibling relationships were derived using UK Biobank provided estimates of pairwise identical by state (IBS) kinships ($>0.5–21^*$IBS0, $<0.7$) and IBS0 ($>0.001$, $<0.008$), the proportion of unshared loci [6]. This approach identified 40,275 siblings from 19,523 families. For the purposes of within-sibship analyses, we restricted the sample to 2 siblings from each family, selecting siblings at random. The analysis sample included 39,046 individuals from 19,523 families.

**Phenotype data**

At baseline, the height of study participants was measured using a Seca 202 device at the assessment centre (field ID: 12144–0.0), body mass index was derived manually from measures of standing height and weight (field ID: 21001.0.0), systolic blood pressure was measured using an automated reading from an Omron Digital blood pressure monitor (field ID: 4080–0.0). Educational attainment was defined as in a previous study [56], using questionnaire data on qualifications to estimate the number of years spent in full-time education (field ID: 6138). Coronary artery disease cases were diagnosed using International Classification of Disease (10th edition) (ICD10) and Operating Procedure System (OPS) codes from either hospital events (Hospital Episode Statistics) or underlying cause of death from the death register. The following ICD10 (I21, I22, I23, I24, I25, Z955) and OPS codes (K40-K46, K471, K49, K50, K75) [57] were used to classify diseased cases. North-south (field ID: 129) and east-west (field ID: 130) birth coordinates were derived from self-reported town of birth.

Alcohol consumption was defined as in a previous study [24]. In brief, participants were asked to estimate their current alcohol intake frequency (daily or almost daily, three or four times a week, once or twice a week, one to three times a month, special occasions only, never,

prefer not to say) (ID: 1558–0.0). Individuals reporting a current intake frequency of at least once or twice a week were asked to estimate their average weekly intake of a range of different alcoholic beverages (red wine, white wine, champagne, beer, cider, spirits, fortified wine) (ID: 1568–0.0, 1578–0.0, 1588–0.0, 1598–0.0, 1608–0.0). We converted intake frequencies to weekly alcohol consumption in units by converting the questionnaire measurements to units: measures for spirits (1 unit), glasses for wines (2 units) and pints for beer/cider (2.5 units). Individuals reporting current intake frequency of "one to three times a month", "special occasions only" or "never" (for whom this phenotype was not collected), were assumed to have a weekly alcohol consumption volume of 0. We removed 189 pairs with outlying values (>5 S.D from the mean) from one or more members.

**Genotyping**

UK Biobank study participants (N = 488,377) were assayed using the UK BiLEVE Axiom Array by Affymetrix1 (N = 49,950) and the UK Biobank Axiom Array (N = 438,427). Directly genotyped variants were pre-phased using SHAPEIT3 [58] and then imputed using Impute4 using the UK10K [59], Haplotype Reference Consortium [60] and 1000 Genomes Phase 3 [61] reference panels. Post-imputation, data were available for approximately ~96 million genetic variants. More detail is contained in previous publications [35,62].

**Genome-wide association studies.** Summary statistics from previous published GWAS, independent from UK Biobank, were used for information on SNPs associated with coronary artery disease [63], body mass index [64], educational attainment [56] and height [65].

Genome-wide summary data were not available for a recent systolic blood pressure GWAS [66], so we performed a GWAS of systolic blood pressure using UK Biobank. To remove sample overlap, we excluded the 47,435 spouse pairs from the analysis and used the remaining sample of 367,963 individuals of self-report European descent. A GWAS was conducted on this sample using a linear mixed model (LMM) association method as implemented in BOLT-LMM (v2.3)[67]. To model population structure in the sample we used 143,006 directly genotyped SNPs obtained after filtering on MAF > 0.01; genotyping rate > 0.015; Hardy-Weinberg equilibrium p-value < 0.0001 and LD pruning to an r2 threshold of 0.1 using PLINK v2.0 [68]. We included the age and sex of participants as covariates in the model.

A set of Genome-wide significant SNPs were generated for each trait by LD clumping relevant summary statistics ($P < 5 \times 10^{-8}$, $r^2 < 0.001$, clumping distance = 10000 kb) using the 1000 Genomes Phase 3 GBR samples [61] as the reference panel. For alcohol consumption, we used a missense variant (rs1229984) in *ADH1B* strongly associated with alcohol behaviour, as in a previous study [24].

## Theory of within-spouse pair comparisons

The phenotype $P$ of individual $I$ can be modelled as a function of independent factors; genetics $G$, the environment $E$, age, sex and a stochastic variance term $\in$.

$$P_I = G_I + E_I + Age_I + Sex_I + \in_I$$

When considering male-female spouse pairs, we can decompose the influence of the environment $E$ on $P$ into effects of the shared environment between spouses $SE$ (e.g. during cohabitation) and effects of the non-shared environment $NSE$. For example, for the male $M$ and female $F$ in pair $K$:

$$P_{KM} = G_{KM} + (SE_K + NSE_{KM}) + Age_{KM} + Sex_{KM} + \in_{KM}$$

$$P_{KF} = G_{KF} + (SE_K + NSE_{KF}) + Age_{KF} + Sex_{KF} + \in_{KF}$$

We then define the WSP model across spouse pairs as:

$$P^* = G^* + E^* + Age^* + Sex^* + \in^*$$

where the differences between the spouses for each factor are included in the model (e.g. for pair $K$, $P_K^* = P_{KM} - P_{KF}$, $G_K^* = G_{KM} - G_{KF}$, $E_K^* = NSE_{KM} - NSE_{KF}$). The shared environmental terms are by definition equal for men and women and drop out of the model.

For the WSP model to generate an unconfounded estimate of the causal effect of $G$ on $P$, we require that the genetic and environmental difference terms in the between-spouse model are independent, i.e. $Corr(G^*, E^*) = 0$. This assumption could be violated by several factors including assortative mating and indirect genetic effects. For example, if parental genotypes influence their offspring phenotype, then the offspring's genotype would be positively correlated with their parental environment.

**Random and non-random mating.** Consider the WSP model applied to three distinct sets of pairs: a) a random set of males and females (non-spouses), b) spouse pairs under random mating (random spouses), and c) spouse-pairs under assortative mating (assorted spouses). In theory, the environmental differences between pairs would decrease with cohabitation and under assortment on environmental factors such as place of birth and socio-economic status:

$$E^{*NonSpouse} > E^{*RandomSpouse} > E^{*AssortedSpouse}$$

Note that as the environmental differences between pairs tends to zero ($E^* \rightarrow 0$), the bias in the estimated association between $P$ and $G$ will also tend to zero ($bias(P \sim G) \rightarrow 0$) even if $G^*$ and $E^*$ are correlated in the WSP model ($Corr(G^*, E^*) \neq 0$) because the pair would be matched for the confounder, suggesting that comparing assorted pairs could reduce the effect of environmental biases.

We define the mechanism by which spouses assort as spousal compatibility $A$, a pairwise measure of the likelihood that two individuals enter a relationship. If several phenotypes influence assortment, then assortative mating can induce collider bias. For example, assortment on a phenotype influenced by genetic and environmental factors could induce spousal correlations in both genetic and environmental determinants of the phenotype, i.e. $Corr(G_{KM}, G_{KF}) > 0$ & $Corr(E_{KM}, E_{KF}) > 0$. It follows that in the WSP model, spousal genetic differences could be inversely associated with spousal environmental differences, i.e. $Corr(G^*, E^*) < 0$.

## Statistical methods

**Simulations. Model A: Within-spouse pair: spousal correlation for confounders**

In model A, an exposure $X$ influences an outcome $Y$ but the relationship is confounded by life-course exposure to an environmental factor $E$ which influences both $X$ and $Y$. We evaluated the effect of spousal correlations for $E$ on the WSP estimates of the effect of $X$ on $Y$.

Spousal correlations for $E$ were generated by simulating $E$ and a spousal assortment measure $A$ such that $Corr(E, A) = C$. Male-female pairs were defined by ordering $A$ such that $A_{M1} \geq A_{M2} \geq .. A_{M1000}$ and $A_{F1} \geq A_{F2} \geq .. A_{F1000}$ and matching respective males and females, i.e. $A_{M1}$ with $A_{F1}$. This matching induces a spousal correlation for $E$ which converges to $C$ as the sample size increases to infinity.

Using 2,000 simulated individuals (1,000 males and 1,000 females), we generated WSP estimates at a range of values of $C$ (0.1, 0.2, 0.3, 0.4, 0.5, 0.6, 0.7, 0.8, 0.9). Code for model (A) can be found at https://github.com/LaurenceHowe/Between-spouse/blob/master/simulations.R.

**Model B: Within-spouse pair: assortative mating and collider bias**

In model B, individuals assort on two independent phenotypes $X_1$ and $X_2$, that also influence an outcome $O$ such that $Y \sim X_1 + X_2 + \in$. We evaluated the effects of assortment on $X_1$ and $X_2$ on the WSP estimate of the effect of $X_1$ on $Y$. We simulated $A$, $X_1$ and $X_2$ such that $Corr(X_1,$

$A) = C_1$ & $Corr(X_2 A) = C_2$. As above, we then defined pairs by ordering $A$ which induces spousal correlations for $X_1$ and $X_2$.

Using 2,000 simulated individuals (1,000 males and 1,000 females), we generated WSP estimates used varying degrees of spousal assortment ($C_i = 0, 0.1, 0.2, 0.3, 0.4, 0.5$: $i \, \epsilon(1,2)$). The WSP regression model is defined as $Y^* \sim X_1^*$ where $Y^* = Y_{KM} - Y_{KF}$ and $X_1^* = X_{1KM} - X_{1KF}$ for each assorted pair. Code for model (B) can be found at https://github.com/LaurenceHowe/Between-spouse/blob/master/simulations.R.

## Empirical analysis in the UK Biobank

**Within-spouse pair: Genetic and phenotypic differences.** We estimated the correlations between spouses for birth coordinates (north-south, east-west) principal components using a linear regression model in R. Given that the regression model includes the same variable from different individuals, the association estimates are approximately equivalent to correlations.

We defined the genotypic differences at a variant for spouse pair $K$ with individuals $A$ and $B$ as:

$$GenotypeDif_K = Genotype_{KA} - Genotype_{KB}$$

WSP effect estimates of each genetic variant on the relevant phenotype of interest (height, body mass index, systolic blood pressure, educational attainment, coronary artery disease or alcohol consumption) were generated using linear or logistic regression. In the context of binary outcomes, the pair were rearranged so that the phenotypic difference could take the value of either 0 or 1 (for logistic regression), with other variables rearranged accordingly. The sex of the reference individual and the age difference between the spouses were included as covariates:

$$PhenotypeDif_K \sim GenotypeDif_K + AgeDif_K + Sex_{KA}$$

where $PhenotypeDif_K = Phenotype_{KA} - Phenotype_{KB}$

and $AgeDif_K = Age_{KA} - Age_{KB}$

Using the models described above, we generated associations using the WSP model with the spouse-pairs. For comparison, we generated 100 distinct datasets of random-male female pairs which were generated by randomly rearranging the 47,435 spouse-pairs and ensuring that pairs were of different sex. We applied the same within-pair models to the random male-female pairs, taking the median effect estimate and standard error for each variant from the 100 random-pair estimates. To compare WSP and random-pair genetic association estimates, we used an inverse-variance weighted (IVW) approach [69,70]. The IVW approach uses summary data to estimate the effect of a polygenic score from the discovery GWAS, where the genetic variants were selected from, on the phenotype in both models. Using betas from the discovery GWAS as "weights" and betas and standard errors from the WSP and random-pair models, the IVW estimates are calculated across N variants as follows.

$$Beta(IVW) = \frac{\sum_1^n \frac{Weight * Beta}{SE^2}}{\sum_1^n \frac{Weight^2}{SE^2}}$$

$$SE(IVW) = \sqrt{\frac{1}{\sum_1^n \frac{Weight^2}{SE^2}}}$$

Shrinkage in genetic associations for each phenotype, defined as the percentage difference between the random pair IVW estimate and the WSP estimate, was calculated using the delta method assuming no covariance between the estimators.

As we investigated only a single genetic variant for alcohol consumption, we were unable to investigate a trend across genetic variants. Instead we tested for a difference between two means for the WSP and median random-pair estimate [71].

Within-sibship birth coordinate correlations, principal component correlations and shrinkage estimates were generated using very similar methods to the spousal analyses [4,6,36]. Unlike the male-female spouse-pairs, siblings can be different sexes, so we included a sex difference term in the regression models for the shrinkage analysis. Within-sibship estimates were compared with random-pair estimates as in the spousal analysis. Shrinkages in genetic associations for each phenotype were estimated as above.

$$PhenotypeDif_K \sim GenotypeDif_K + AgeDif_K + SexDif_K$$

$$\text{where } SexDif_K = Sex_{KA} - Sex_{KB}$$

We investigated heterogeneity between WSP and within-sibship shrinkage estimates using the difference for two means test [71] assuming no covariance.

As a sensitivity analysis, we included an analysis adjusting for principal components in the random-pair samples to account for population structure differences. We included differences for the first 10 principal components in the random pair models as below. Principal component differences were not included in the WSP or within-sibship models.

$$PhenotypeDif_K \sim GenotypeDif_K + AgeDif_K + SexDif_K + PC1Dif_K.. + PC10Dif_K$$

$$\text{where } PC1Dif_K = PC1_{KA} - PC1_{KB}$$

For comparison, we also considered within-sibship shrinkage estimates from a recent within-sibship GWAS preprint [5]. This preprint reported shrinkage for genetic variants at genome-wide significance ($5\times10^{-8}$) and a more liberal threshold ($1\times10^{-5}$) for height, BMI, educational attainment, SBP and alcohol consumption. Coronary heart disease was not analysed in this study. As the shrinkage estimates were broadly similar between the two thresholds for the 5 phenotypes in this preprint, we considered the shrinkage estimates from the liberal threshold.

**Within-spouse pair: age, SBP and CAD.** The WSP effect estimates of age on CAD and SBP were estimated using the following regression model (linear or logistic dependent on the outcome of interest), including sex of the reference individual and the age difference between-spouses as covariates:

$$PhenotypeDif_K \sim AgeDif_K + Sex_{KA}$$

As above, we repeated analyses using the datasets of random male-female pairs, reporting the median effect size and standard error across the 100 simulated datasets for each model.

## Supporting information

**S1 Table. Model 1: Spousal correlations controlling for confounding.** Results from simulation analyses investigating how the WSP model can control for confounding if spouses assort on the confounder.
(DOCX)

**S2 Table. Model 2: Assortment and collider bias.** Results from simulation analyses investigating how the WSP model may be susceptible to collider bias induced by spousal assortment.
(DOCX)

**S3 Table. Characteristics of the spouse sample (N≤94,870).** A table containing summary-level phenotype information on the characteristics of the UK Biobank spouses, stratified by sex.
(DOCX)

**S4 Table. Spouse and sibling pair correlations for birth coordinates and principal components.** A table containing within-pair correlations for spouses and siblings for north-south and east-west birth coordinates as well as the first 10 principal components.
(DOCX)

**S5 Table. Comparisons of WSP and within-sibship shrinkage estimates.** A table containing WSP and within-sibship shrinkage estimates from this study for height, educational attainment, BMI, SBP and alcohol consumption as well as within-sibship shrinkage estimates from an external preprint.
(DOCX)

**S1 Fig. Simulation results for Within spouse-pair models.** A–Simulations for model (A): Spousal correlations controlling for confounding. As the strength of spousal assortment (spousal correlation) on the confounder ($E$) increases, the within-spouse pair (WSP) estimate of $X$ on $Y$ unadjusted for $E$ (in blue) moves from the confounded unadjusted estimate of 0.45 to the unbiased estimate of 0.30. B–Simulations for model (B): Within spouse-pair: assortment and collider bias. Spousal assortment can induce collider bias in WSP estimates. If spouses assort on two phenotypes $X_1$ and $X_2$ which both affect outcome $Y$, then the association of $X_1$ and $Y$ (or $X_2$ and $Y$) estimated from the WSP model is a biased estimate of the causal effect of $X_1$ on $Y$ (or $X_2$ on $Y$). This bias monotonically increases in the degree of assortment on either $X_1$ or $X_2$.
(PNG)

**S2 Fig. Height of UK Biobank spouse pairs.** Scatter plot showing male spouse height on the X axis and female spouse height on the Y axis for each spouse-pair.
(PNG)

**S3 Fig. BMI of UK Biobank spouse pairs.** Scatter plot showing male spouse BMI on the X axis and female spouse BMI on the Y axis for each spouse-pair.
(PNG)

**S4 Fig. SBP of UK Biobank spouse pairs.** Scatter plot showing male spouse SBP on the X axis and female spouse SBP on the Y axis for each spouse-pair.
(PNG)

## Author Contributions

**Conceptualization:** Laurence J. Howe, George Davey Smith.

**Data curation:** Laurence J. Howe.

**Formal analysis:** Laurence J. Howe, Thomas Battram.

**Funding acquisition:** Neil M. Davies, George Davey Smith.

**Investigation:** Laurence J. Howe, Thomas Battram, Tim T. Morris, Fernando P. Hartwig, Gibran Hemani, Neil M. Davies, George Davey Smith.

**Methodology:** Laurence J. Howe.

**Supervision:** Gibran Hemani, Neil M. Davies, George Davey Smith.

**Writing – original draft:** Laurence J. Howe.

**Writing – review & editing:** Laurence J. Howe, Thomas Battram, Tim T. Morris, Fernando P. Hartwig, Gibran Hemani, Neil M. Davies, George Davey Smith.

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
