## [Decision Letter · Decision Letter 0]

25 Mar 2021

Dear Dr Howe,

Thank you very much for submitting your Research Article entitled 'Assortative mating and within-spouse pair comparisons' to PLOS Genetics.

The manuscript was fully evaluated at the editorial level and by independent peer reviewers. The reviewers appreciated the attention to an important problem, but raised some substantial concerns about the current manuscript. Based on the reviews, we will not be able to accept this version of the manuscript, but we would be willing to review a much-revised version. We cannot, of course, promise publication at that time.

If you decide to revise the manuscript for further consideration at PLOS Genetics, please aim to resubmit within the next 60 days, unless it will take extra time to address the concerns of the reviewers, in which case we would appreciate an expected resubmission date by email to plosgenetics@plos.org.

[LINK]

We are sorry that we cannot be more positive about your manuscript at this stage. Please do not hesitate to contact us if you have any concerns or questions.

Yours sincerely,

Samuli Ripatti

Associate Editor

PLOS Genetics

David Balding

Section Editor: Methods

PLOS Genetics

As you can see from the reviewers comments, they found the work interesting and potentially important, but what would be particularly needed is to make a better fit to a broader genetics audience. In its current format, the paper would make a better fit to an epidemiological journal. However the reviewers give several suggestions on how to modify it to be better suited to a genetics audience with better examples from genetic analyses and examination of the behaviour of the model across a range of circumstances in simulated and/or empirical data.

Reviewer's Responses to Questions

**Comments to the Authors:**

Reviewer #1: Review been attached

Reviewer #2: This study presents a comparison of two different “within-family” models for testing genetic associations. The concept is clear that a within-spouse pair analysis may account for shared environment as the shared environment among spouses is the same. However, I have a few concerns:

1. I do not see how a WSP model controls for ancestry confounding if spousal differences correlate with their ancestry differences? Yes, pairs may be matched for the home environment, but a correlation between delta_phenotype and delta_genotype could certainly be driven by delta_ancestry among spousal pairs, perhaps even more so as all other environmental similarity may be controlled for. If one fits the model in a mixed model framework does it help control for stratification? What if there is a phenotypic correlation with PC1 and mating is random with respect to PC1, is there covariance between the effect size estimates and the PC loadings. The sibling pair analysis, due to mendelian segregation should be unbiased of this, but I do not see a strong consideration of genetic stratification effects in this work, which I feel leaves it lacking.

2. Why use summary statistics from previously published non-UKB meta-analyses? Why not compare to directly estimating the effects using a mixed-effects association model in the UK Biobank? Non-UKB meta-analysis estimates are obtained from very heterogeneous cohorts in the worst type of association study methodology (single-marker marginal associations with some arbitrarily determine dnumber of PCs). The current comparison is valid, but I think a comparison of analysing the same data in a different way is also warranted. Note systolic blood pressure was analysed in this way (within UKB analysis) and the shrinkage, especially from the spouse pair model is much less than for some of the other traits.

3. I would also suggest rather than taking the most statistically significant estimates, which acertains a very specific subset of variants with specific LD and MAF properties, that comparisons are done across a range of markers. Perhaps prune the data for LD beforehand and analyse all the markers and then observe the shrinkage genome-wide and as a function of the association test statistic values?

Reviewer #3: Please see attachment

**Have all data underlying the figures and results presented in the manuscript been provided?**

Reviewer #1: Yes

Reviewer #2: Yes

Reviewer #3: Yes

PLOS authors have the option to publish the peer review history of their article (what does this mean?). If published, this will include your full peer review and any attached files.

Reviewer #1: No

Reviewer #2: No

Reviewer #3: No

---

## [Decision Letter · Decision Letter 1]

11 Aug 2021

Dear Dr Howe,

Thank you very much for submitting your Research Article entitled 'Assortative mating and within-spouse pair comparisons' to PLOS Genetics.

The manuscript was fully evaluated at the editorial level and by independent peer reviewers.  Both reviewers find that the revised article has improved considerably and has substantial merit, but Reviewer 3 raises a fundamental question about the role of collider bias in the WSP design and how it impacts the control of confounding in an empirical setting.  The editors would like to see these addressed. The reviewer suggest two possible ways to quantify the problem. Either (or both) of these additional analyses would potentially strengthen the paper and help drive home its main argument.  Alternatively the authors may find a better approach to address Reviewer 3's concerns.

Therefore we are sending it back to you for major revision, but this time with a narrower focus on the issue to be addressed and we look forward to receiving a revised submission.  As usual, please detail responses to the reviewers in a letter and describe the changes you have made in the manuscript.

If you decide to revise the manuscript for further consideration at PLOS Genetics, please aim to resubmit within the next 60 days, unless it will take extra time to address the concerns of the reviewers, in which case we would appreciate an expected resubmission date by email to plosgenetics@plos.org.

[LINK]

We are sorry that we cannot be more positive about your manuscript at this stage. Please do not hesitate to contact us if you have any concerns or questions.

Yours sincerely,

Samuli Ripatti

Associate Editor

PLOS Genetics

David Balding

Section Editor: Methods

PLOS Genetics

Reviewer's Responses to Questions

Reviewer #2: I believe that the authors have argued for the approach taken in this manuscript very well and in doing so have alleviated my concerns sufficiently for meta support publication. While I think more work needs to be done on this topic, what is presented here is very interesting and represents an ideal starting point for future research.

Reviewer #3: I appreciate the authors significantly reworked the presentation of the manuscript, which in my view is now much easier to follow. I think both the Introduction and Discussion sections are now much improved.

My main concern is the following. The manuscript strives to drive home two points (based on the abstract): (1) WSP design can reduce confounding, but (2) WSP design is susceptible to collider bias. Both confounding and collider bias are addressed in simulation, but there is no direct evidence of collider bias is at play in empirical data analysis. Rather, the authors inferred that collider bias is happening because shrinkage of the effect size estimates in WSP design is larger than within-sibship design; since within-sibship design is perfectly controlling for confounding due to ancestry/demography/family, the additional shrinkage is attributed to collider bias that induces a negative correlation. But ultimately, this is still only an indirect inference by deduction. Collider bias is thus, in my view, still not strongly supported by empirical data analysis. Would some approach like that published in Day et al., AJHG 2016 (PMID 26849114) be feasible in this setting to provide a direct demonstration of collider bias?

In fact, by the author’s argument, shrinkage in the WSP design need not be greater than that in the within-sibship design for collider bias to be at play, since the correlation between spouses in principal component space is quite a bit less than the correlation between sibs, suggesting that WSP design is much less efficacious to control for confounding (at least due to ancestry), than the within-sibship design. Yet, it is not clear the degree to which WSP design could be controlling for confounding, while additional shrinkage could be attributed to collider bias. So some form of partitioning of the effects attributed to correction for confounding could contribute to evidence that collider bias is operating (though still indirectly).

Lastly, a slightly minor concern, is that the authors assumed that spouses assort on ancestry compared to non-spousal pairs (e.g. paragraph 2 of Discussion). I think this comes from the belief that ancestry are the most common confounder in genetic association studies, and there are correlations in PC space between spouses – but it is not actually demonstrated that the correlation in PC space is related to ancestry, considering that only the white British UKB individuals were used in the analysis. (And note that PCA would be sensitive to the choice of individuals used in the analysis, thus subject to all upstream ascertainment practices. In other words, demonstration would need to occur in-analysis, rather than citing publications investigating population structure in UKB in general).

**Have all data underlying the figures and results presented in the manuscript been provided?**

Reviewer #2: Yes

Reviewer #3: Yes

PLOS authors have the option to publish the peer review history of their article (what does this mean?). If published, this will include your full peer review and any attached files.

Reviewer #2: No

Reviewer #3: No

---

## [Editor Report · Decision Letter 2]

15 Oct 2021

Dear Dr Howe,

We are pleased to inform you that your manuscript entitled "Assortative mating and within-spouse pair comparisons" has been editorially accepted for publication in PLOS Genetics. Congratulations!

Yours sincerely,

Samuli Ripatti

Associate Editor

PLOS Genetics

David Balding

Section Editor: Methods

PLOS Genetics

**Data Deposition**

http://datadryad.org/submit?journalID=pgenetics&manu=PGENETICS-D-21-00140R2

**Press Queries**

---

## [Editor Report · Acceptance letter]

29 Oct 2021

PGENETICS-D-21-00140R2 

Assortative mating and within-spouse pair comparisons 

Dear Dr Howe, 

We are pleased to inform you that your manuscript entitled "Assortative mating and within-spouse pair comparisons" has been formally accepted for publication in PLOS Genetics! Your manuscript is now with our production department and you will be notified of the publication date in due course.

With kind regards,

Agnes Pap

PLOS Genetics

On behalf of:
